# A Rare Cause of Deep Vein Thrombosis in a Young Orchestra Conductor

**DOI:** 10.3390/diagnostics14040354

**Published:** 2024-02-06

**Authors:** Anca Mihaela Lungu, Irina Mariella Andrei, Gabriela Uscoiu, Mihai Grigore, Adriana Mihaela Iliesiu

**Affiliations:** 1Cardiology Department, “Prof. Dr. Th. Burghele”, 050659 Bucharest, Romania; anca-mihaela.lungu@rez.umfcd.ro (A.M.L.); gabriela.uscoiu@umfcd.ro (G.U.); mihai.grigore@drd.umfcd.ro (M.G.); adriana.iliesiu@umfcd.ro (A.M.I.); 2Internal Medicine and Cardiology Department, Carol Davila University of Medicine and Pharmacy, 020021 Bucharest, Romania

**Keywords:** effort-induced upper extremity deep vein thrombosis, thoracic outlet syndrome, anticoagulants, surgical decompression

## Abstract

Upper extremity deep vein thrombosis (DVT) of the axillary/subclavian veins is rare (5–10% of DVT). After clinical suspicion and duplex ultrasound, anticoagulation, surgical decompression and sometimes thrombolysis are mandatory due to complications. We discuss the case of a young healthy orchestra conductor with primary DVT of the left upper extremity and concomitant left shoulder musculo-tendinous traumatic injury. Symptoms of both conditions and subtle signs of upper extremity DVT delayed the diagnosis until full-blown DVT occurred. After successful anticoagulation and surgical TOS (thoracic outlet syndrome) decompression, evolution was favorable, without recurrent thrombosis.

Upper extremity deep vein thrombosis (DVT) is rare, with an incidence of 1–2:100,000 per year [1]. DVT commonly involves subclavian and axillary veins and in most cases, it is secondary to the use of central venous catheter or pacemakers or is associated with malignancy or thrombophilia. The primary upper extremity DVT is the venous form of thoracic outlet syndrome (vTOS), accounting for 3% of all cases and may appear in the setting of congenital (e.g., cervical ribs) or acquired (e.g., muscle hypertrophy) anatomic abnormalities. vTOS typically occurs in young, otherwise healthy persons, as sudden or progressive pain and swelling in the upper extremity following strenuous and vigorous upper extremity activity, being also named "effort" thrombosis or Paget–Schroetter Syndrome [1].

A therapeutic approach with anticoagulation, catheter-directed thrombolysis in selected cases, and thoracic outlet decompression is mandatory for relieving symptoms and preventing complications.

A 32-year-old young orchestra conductor, a healthy male, was admitted to our department with pain in the neck, left shoulder and arm associated with arm swelling, cyanosis and dilated collateral veins over the proximal upper limb and upper chest. The high clinical suspicion of upper limb DVT was confirmed by venous duplex ultrasound, revealing occlusive thrombus in the left subclavian and axillary veins with extension into brachial veins, as shown in Figure 1. Exclusion of thrombophilia and occult malignancy supported the final diagnosis of primary, effort-related, subacute (beyond two weeks) left DVT of the axilosubclavian and brachial veins, the venous form of TOS. Due to the late diagnosis of vTOS, oral anticoagulation with a direct oral anticoagulant (DOAC), Apixaban 5 mg bid, was initiated.

Three weeks before, the patient had been admitted as an outpatient to an emergency hospital with mild pain in the upper left thorax, shoulder and arm, with no noticeable arm swelling. The ECG and chest X-ray had been normal and pulmonary embolism had been excluded by computed tomography (CT). The magnetic resonance imaging (MRI) had shown post-traumatic (secondary to repetitive arm movements) muscle and tendon injuries of the left shoulder. An orthopaedic evaluation had been recommended.

After establishing the diagnosis of thrombosis, the patient was further evaluated by the thoracic surgery team and CT angiography showed left subclavian vein compression at the level of first rib, as shown in Figure 2.

After three months of oral anticoagulation, thoracic outlet decompression with first rib resection through a trans-axillary approach was performed successfully. Oral anticoagulation was continued for another 3 months after surgery and the patient became asymptomatic. During ultrasound reevaluation, the patency of the veins was complete, without thrombus, spontaneous contrast, venous stenosis or recurrent DVT, as shown in Figure 3. Shortly after surgical decompression, the patient resumed his professional activity. At the 6- and 12-month follow-up, the clinical examination and venous ultrasound were normal.

Upper extremity spontaneous DVT is a relatively rare form of TOS. A unique aspect is that this patient did not display any neurological symptoms upon presentation, with the neurological form being the most prevalent. It is a known fact that TOS commonly affects individuals performing high-intensity repetitive movements involving the upper extremity, most often in athletes. However, the incidence of symptomatic TOS is also greater in musicians than in general population, as a consequence of repetitive movements [2].

When there is a high clinical suspicion of upper extremity DVT, diagnosis is confirmed with venous ultrasound, showing dilated veins, partially compressible or incompressible thrombotic material and absence of venous flow in cases of obstructive thrombus. It is non-invasive, inexpensive and has a sensitivity of 97% and a specificity of 96% [3]. CT and MRI are ccomplementary investigations, more likely used for a better evaluation of local anatomy. The patient was not sent for venous ultrasound until later in our department, when the clinical picture was highly suggestive for upper extremity DVT.

The aim of upper extremity DVT therapy is venous recanalization and recurrence prevention. Without treatment, the complications are the same as for lower limb DVT, recurrence of thrombosis, post-thrombotic syndrome and pulmonary embolism. In patients with acute upper extremity DVT and severe symptoms (important pain, edema, functional impairment, dyspnea or thoracic pain), subclavian or axillary vein thrombosis, first line treatment is catheter guided or systemic thrombolysis followed by anticoagulant treatment [4]. Anticoagulation and surgical correction of the TOS are the mainstay of upper extremity DVT therapy. There are no RCTs on the use of thrombolysis in patients with upper extremity DVT. In the CHEST guidelines on treatment of VTE, thrombolysis associated with anticoagulant therapy has a 2C class indication [4].

The general consensus is that patients should receive anticoagulant treatment for at least 3 months, either with low-molecular-weight heparin, vitamin K antagonists (VKAs) or DOACs.

A recent trial that included 61 patients with a first episode of upper extremity DVT treated with either Apixaban 10 mg bid for 7 days, followed by 5 mg bid or Rivaroxaban 15 mg bid for 21 days, followed by 20 mg od, showed good efficacy of DOACs, with no recurrence during treatment or significant bleeding and partial or complete recanalization after at least 3 months of treatment. The same data of safety and efficiency (no significant bleeding and partial or complete recanalization) were reported by a Swedish study of 55 patients treated with DOAC for 3 to 6 months [5]. The latest guidelines on DVT recommend DOACs over VKAs with a 2B class of indication [4]. These studies included patients with acute DVT; therefore, there are no data to support the use of Apixaban 10 mg dose bid for 7 days, followed by 5 mg bid or Rivaroxaban 15 mg bid for 21 days, followed by 20 mg od in subacute thrombosis.

The minimally invasive surgical correction consists of first rib and costal-clavicular ligament resection through a trans-axillary approach. For those who require correction surgery, anticoagulant treatment must be continued at least 3 months after surgery. Stent placement is contraindicated because of the high risk of stent fracture during arm vigorous movements [6]. In the absence of local anatomy correction, recurrent thrombosis is frequent.

A point of discussion was the timing of surgery and the optimal duration of anticoagulant treatment after decompression surgery, taking into consideration the limited data from the literature. Some centers recommend surgery right after thrombolysis or anticoagulant initiation and others recommend at least three months of anticoagulant treatment, with good results in both cases. In our patient, the surgical correction was performed after three months of oral anticoagulation, followed by another three months of anticoagulation with uneventful evolution.

What made this case remarkable was the association of two diseases with overlapping symptoms in the beginning, leading to delayed diagnosis and treatment of the upper extremity DVT. The multidisciplinary, medical and surgical, therapeutic approach was curative, allowing the patient’s resumption of the orchestra conductor activity, without DVT recurrence.

## Figures and Tables

**Figure 1 diagnostics-14-00354-f001:**
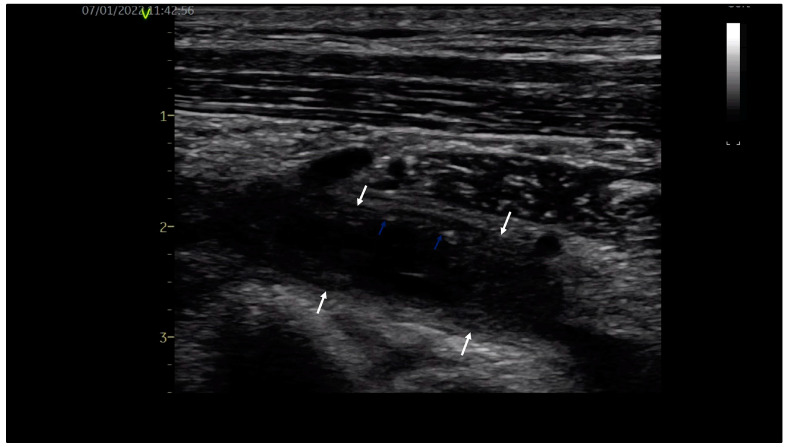
Venous ultrasound: subacute left subclavian vein thrombosis with hypoechoic thrombotic material (white arrows) and small hyperechoic areas (blue arrows).

**Figure 2 diagnostics-14-00354-f002:**
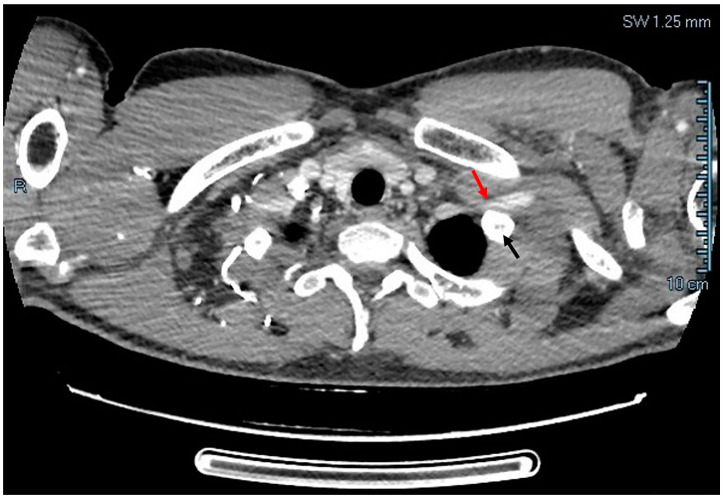
CT angiography: left subclavian vein compressed (red arrow) by first rib (black arrow).

**Figure 3 diagnostics-14-00354-f003:**
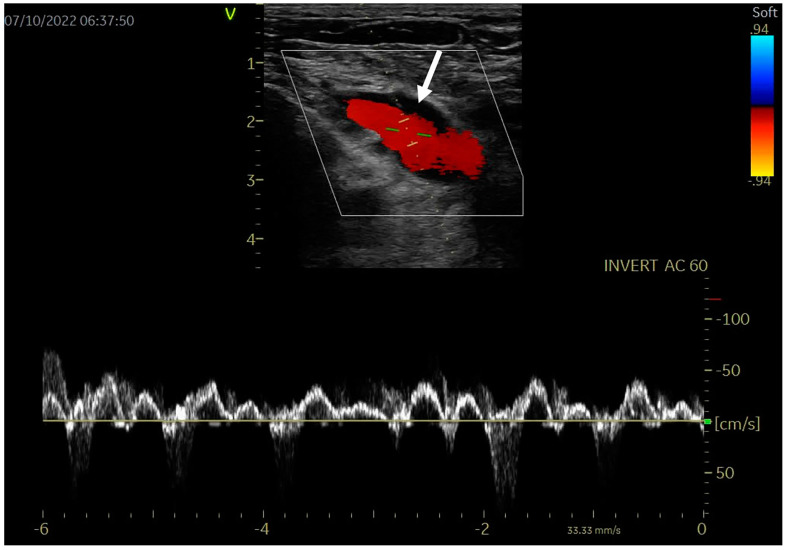
Venous ultrasound: left subclavian vein recanalization (white arrow), with normal color Doppler (red content) and pulsed Doppler flow.

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
