# Peer review of "A Rare Cause of Deep Vein Thrombosis in a Young Orchestra Conductor"

_diagnostics, 2024, doi:10.3390/diagnostics14040354_

Round 1

Reviewer 1 Report

Comments and Suggestions for Authors

DVT secondary to Paget Schroetter are not so rare and the title should be modified according ; title could integrate patient job which is more rare

Figure 1. This is very difficult to see anomalies pointed ; I am not sure that this figure is essential

This is strange to start the manuscript with possible treatments. In fact, text in the figure should be not place here instead of in the principal draft but at the 2 places.

Do you consider that the initial diagnostic of muscle/tendon injury was false ? Which was the traumatism implied ?

What is the link between late diagnostic and the anticoagulant treatment? This is to explain why patient was not treated with 10 mg bid the first 7 days ?

How long was the follow up of the patient after the treatment to conclude to the absence of recurrence ?

What do you consider as severe symptom needing thrombolysis ?

Author Response

Thank you very much for taking the time to review this manuscript. Please find the detailed responses below and the corresponding revisions/corrections highlighted in the re-submitted files

  1. DVT secondary to Paget Schroetter are not so rare and the title should be modified according ; title could integrate patient job which is more rare.

Thank you for this suggestion. We agree that it is important to point this particularity in the title. In consequence, we change the title to “A rare cause of deep vein thrombosis in a young orchestra conductor”

  1. Figure 1. This is very difficult to see anomalies pointed ; I am not sure that this figure is essential.

Thank you for your observation. We consider this figure suggestive for subacute thrombosis diagnosis and we consider this important for the paper.

  1. This is strange to start the manuscript with possible treatments. In fact, text in the figure should be not place here instead of in the principal draft but at the 2 places.

Thank you for this comment. We agree that the text figure belongs to principal draft. We changed it, along with the text in the other 2 pictures.

  1. Do you consider that the initial diagnostic of muscle/tendon injury was false ? Which was the traumatism implied ?

We consider that there was an association of two diseases with overlapping symptoms – muscle/tendon injury secondary to repetitive arm movements and upper extremity deep vein thrombosis. We added a new paragraph at the end of the manuscript to point out this aspect. The post-traumatic lesions are considered in context of repetitive arm movements.

  1. What is the link between late diagnostic and the anticoagulant treatment? This is to explain why patient was not treated with 10 mg bid the first 7 days ?

Yes. We decided to treat the pacient with apixaban 5 mg bid, given the fact that 10 mg bid is recommended for the first 7 days in cases of acute thrombosis. We added this information in the manuscript.

  1. How long was the follow up of the patient after the treatment to conclude to the absence of recurrence?

The patient was evaluated 6 and 12 months after surgical intervention without signs of recurrence. Given the fact that the mechanical compression, the underlying cause of thrombosis, was removed, we consider recurrence improbable.

  1. What do you consider as severe symptom needing thrombolysis?

We consider severe symptoms – important pain, edema, functional impairment, dyspnea or thoracic pain. We mentioned them in the text also.

Reviewer 2 Report

Comments and Suggestions for Authors

General comments: 

The authors describe an uncommon case of DVT in the subclavian vein after continuous effort in a young orchestra conductor.  

Specific comments: 

Since the case reported is rare some other medical history and laboratory assessment is recommended including D Dimer analysis and thrombophilia history and evaluation.

Author Response

Thank you very much for taking the time to review this manuscript. Please find the detailed responses below and the corresponding revisions/corrections highlighted in the re-submitted files.

  1. Since the case reported is rare some other medical history and laboratory assessment is recommended including D Dimer analysis and thrombophilia history and evaluation.

Thnak you for this suggestion. Given the presentation in our clinic highly suggestive for thrombosis, we didn’t consider necessary to evaluate D-dimers level as an elevated level is only suggestive, but not necessary for diagnostic. The patient was evaluated for thrombophilia and it was excluded – we added this information in the manuscript.